# Trends and Factors Associated with Obesity Prevalence in Rural Australian Adults—Comparative Analysis of the Crossroads Studies in Victoria over 15 Years

**DOI:** 10.3390/nu14214557

**Published:** 2022-10-28

**Authors:** Stephanie Hannah, Kingsley E. Agho, Milan K. Piya, Kristen Glenister, Lisa Bourke, Uchechukwu L. Osuagwu, David Simmons

**Affiliations:** 1School of Science, Western Sydney University, Campbelltown, NSW 2560, Australia; 2School of Health Sciences, Western Sydney University, Campbelltown, NSW 2560, Australia; 3Macarthur Diabetes Endocrinology Metabolism Services, Camden and Campbelltown Hospitals, Campbelltown, NSW 2560, Australia; 4School of Medicine, Western Sydney University, Campbelltown, NSW 2560, Australia; 5Department of Rural Health, University of Melbourne, Wangaratta, VIC 3010, Australia; 6Department of Rural Health, University of Melbourne, Shepparton, VIC 3632, Australia; 7Bathurst Rural Clinical School (BRCS), School of Medicine, Western Sydney University, Bathurst, NSW 2795, Australia

**Keywords:** obesity, prevalence, crossroads, rural health, australia, victoria

## Abstract

This study examined the changes in the prevalence of obesity and associated lifestyle factors using data from repeated cross-sectional, self-reported surveys (Crossroads I: 2001–2003 and Crossroads II: 2016–2018, studies) and clinic anthropometric measurements collected from regional and rural towns in the Goulburn Valley, Victoria. Given that past community studies have only focused categorically on dietary intake, or assessed caloric energy intake, we examined the difference in broad dietary practices at two different times. Clinical assessments from randomly selected household participants aged ≥18 years were analyzed. Differences in obesity prevalence were calculated for each individual variable. Logistic regression was used to determine the odds ratios (95% confidence intervals (CI)) with and without adjustment for key lifestyle factors. There were 5258 participants in Crossroads I and 2649 in Crossroads II surveys. Obesity prevalence increased from 28.2% to 30.8% over 15 years, more among those who ate fried food, but decreased significantly among rural dwellers (31.7: 27.0, 36.8% versus 25.1: 22.9, 27.5%) and those who had adequate fruit intake (28.5: 25.0, 32.3% to 23.9: 21.8, 26.2%). Obesity was associated with older age (≥35 years), use of fat-based spreads for bread (adjusted odds ratio, aOR:1.26: 1.07, 1.48) and physical inactivity. The increase in obesity prevalence especially in the rural towns, was associated with unhealthy dietary behaviour which persisted over 15 years. Understanding and addressing the upstream determinants of dietary intake and choices would assist in the development of future health promotion Programs.

## 1. Introduction

Obesity is recognized as a global epidemic [1] and the most neglected public health problem in the modern world [1], with approximately 650 million people across both developing and developed nations reported to be living with obesity [2]. Body mass index (BMI) is used as a measure of general obesity, especially in epidemiological studies, while waist circumference (WC) can determine central obesity [3]. Both general and central obesity significantly increase the risk of various chronic diseases, with one study [4] finding healthy life years in adults with severe obesity could decrease by 25% [4]. In Australia, the 2018 National Health Survey found that approximately 12.5 million adults were considered to have overweight or obesity, accounting for 67% of the population [5]. If obesity continues to follow the current rising trend, this figure is estimated to exceed 18 million by 2030 [6]. 

The Crossroads study holds a unique position in that it presents a comprehensive, comparative health data set of a rural Australian community. Using the Crossroads data, one study specifically investigated how the availability of takeaway and restaurant options impacted on the prevalence of adult obesity in rural Victorian communities. The study found no significant relationship between obesity prevalence and takeaway consumption in the Goulburn Valley [7]. These data suggest that the dietary behaviors of the community that are influencing obesity prevalence, and its subsequent health complications, may be existing in food consumption within the home. A comparative analysis of the Crossroads I and II data will revisit such dietary findings and determine if trends are evident in the community’s food choices. While the relationship between obesity and dietary consumption has been well investigated by other researchers in specific rural and regional populations [7,8,9,10,11,12,13] as shown in the Appendix A, no other study was found to offer a comparative analysis over time. Studies undertaken to measure the quality of dietary intake, however varied in population size and demographic characteristics, commonly concluded that the quality of dietary intake has significant room for improvement.

The aim of this study was to determine factors that lead to changes in obesity prevalence and food consumption over 15 years within rural Australian towns using the Crossroads repeated cross-sectional data. Using a comparative analysis of the two data sets, this study examined the changes in food choices, lifestyle, demographics, and obesity prevalence within these rural communities. The findings of this study will provide insight into the factors that influence obesity prevalence within this rural Victorian community. In addition, the findings from this investigation hold the potential to guide future health policy development to promote positive diet and other lifestyle changes, particularly within regional and rural communities. It may also serve to steer the direction of future research within the field of obesity and dietary intake patterns. 

## 2. Materials and Methods

### 2.1. Setting

Victoria, in South-Eastern Australia, is currently home to 6.64 million people and is the second most densely populated state in Australia [9]. In 2018 the Australian Bureau of Statistics reported that 31.8% of Victorian residents were living with obesity [5], showing steady growth in combined overweight and obesity prevalence of the state from 45.4% in 2002 [10] to 68.3% in 2018, with residents of disadvantaged areas being disproportionately affected [5]. The Goulburn Valley, which encompasses the Shepparton region of rural Victoria, is an economically and culturally diverse community with a population of 129,971 people recorded in the 2016 Census [11]. In 2001 the region was identified as having a gross shortage of medical and health professionals, which was believed to be a contributing factor to a higher-than-average standardized mortality rate for the state of Victoria [12]. The rising number of health problems linked to a population with increasing obesity would further increase the demand and use of health care services [13]. The region was chosen for its poor health outcomes and limited health resources at the time [8].

### 2.2. Study Design

This was a repeated cross-sectional population study. Data used for this study were from the 2001–2003 and 2016–2018 Crossroads household surveys [7,8,12] and clinical measures. The data were used to examine the trends in obesity prevalence, and to examine the factors associated with obesity in Victoria. Examining the predictors of obesity, the two-survey data were pooled. The Crossroads studies provide information on a wide range of socio-economic, demographic, diet, lifestyle, and health characteristics (including knowledge and attitudes). Sampling techniques utilized in obtaining the information and the random recruitment of households have been discussed in detail elsewhere [12]. The survey data was collected using REDCap electronic data capture tools (Research Electronic Data Capture, Vanderbilt University, Nashville, TN, USA) in Crossroads II [12]. Crossroads I survey responses were taken on paper, then entered into a database (Microsoft Access, 2002, v 10.0). 

The household survey comprised of 90 questions relating to demography, lifestyle, general health, and health care which would be self-reported by the participant. Surveys were conducted face to face by a member of the research team and responses recorded in person [7,8]. Supplementary questionnaires were provided to individuals who answered as having certain chronic illnesses in the household survey. Clinic participants were also given a range of surveys to complete, including gender specific questions, diet, lifestyle, health knowledge and attitudes. In total, there were nine surveys designed to collect comprehensive data from the cohort. Questionnaires remained largely the same across both studies for consistency and accuracy in data analysis and the merged dataset (*n* = 7907) was used in the analyses of factors associated with change in obesity prevalence over the two years period.

### 2.3. Participants

Participants were only those living in the Goulburn Valley region, Victoria, at the time of the study. Survey participants included all residents of randomly selected household aged 16 years or older who had lived in the region for a minimum of six months. One adult from each household (aged at least 18 years) was randomly selected to undergo physical examination at a screening clinic, as has previously been described [8,12]. People who were pregnant were excluded from participating in the clinical assessment during the Crossroad studies, due to the physiological changes during pregnancy and the inconsistency of clinical anthropometric measurements. 

Overall, 70.3% (*n* = 5258) of the invited households in Crossroads I responded to the survey and 61.3% of respondents attended the clinical assessment [7]. In Crossroads II, the response rate was 62.7% (*n* = 2649) out of the 3022 households that were invited to participate in the survey, with 48.1% attending the clinical assessment [8,12]. The participants were from households in the regional center of Shepparton and Mooroopna and the surrounding six towns in the Crossroads I [12]. In Crossroads II, the number of surveyed rural towns was reduced to three of the original six for pragmatic/financial reasons but still included the regional center towns of Shepparton and Mooroopna.

### 2.4. Clinical Measures

Clinic attendees participated in a range of tests to assess biological markers, via blood tests, urine tests, oral glucose tolerance tests and anthropometric measurements [12]. These tests helped to achieve an accurate clinical assessment of the individual at that particular point in time. The study used the World Health Organization (WHO) criteria for diagnosing obesity, being a BMI ≥ 30 kg/m^2^ indicating general obesity, and a WC of ≥94 cm (≥37 inches) in males and ≥ 80 cm (≥31 inches) in females to determine central obesity [1,8,14]. Weight and height measurements used Seca 813 electronic weight scales (Seca, Hamburg, Germany) and Seca 213 portable stadiometer (Seca, Hamburg, Germany) [12]. Waist measurements were taken using a medical grade measuring tape from Med Shop, Melbourne Australia [12]. Waist measurements were taken from the half-way mark between the bottom of the rib cage and the top of the iliac crest. All anthropometric measures were taken three times, with the closest two recordings being averaged for a final result [12]. Participants were also asked to self-report data on exercise, diet, smoking, alcohol consumption, among other lifestyle related questions. All clinical measures were taken in the same format across both studies for consistency and data integrity in comparisons [12].

### 2.5. Outcome and Exploratory Variables

The key outcome variable in the study was obesity prevalence in surveyed towns (as calculated from weight and height measured at the clinic), which was categorized in binary form as a “Yes” (1 = if the participant had obesity [1,8,14]) and otherwise “No” (0). Previous studies on obesity prevalence, especially from rural areas [7,8,15,16,17], played a role in the exploratory variables selected for the study based on the data available in the pooled dataset. These variables were grouped into five classes: The demographic level factors included age group, gender, ethnicity (Australia born versus non-Australia born) and place of residence (rural or regional).The socio-economic level factors considered were participant’s working status, educational status and marital status which measures the economic status of participants in this study. Due to a lack of data in Crossroads II, income variable was not included.Dietary level factors consisting of the frequency of vegetable intake (classified as adequate, i.e., having the recommended 5 serves of vegetable per day, otherwise inadequate), fruit serves (classified as adequate, i.e., having the recommended 2 serves of fruit per day, otherwise, inadequate), and dairy intake (classified as adequate, i.e., having the recommended 2 serves of dairy per day, otherwise, inadequate), as well as consumption of takeaways and fat-based spreads, which included the use of butter, olive oil, margarine or dripping (all categorized as Yes/No).Cooking method factors were considered including how meat, egg and chicken were usually cooked in participants homes.Lifestyle factors were also assessed which consisted of smoking status (previous and active smokers versus never smoked), alcohol consumption (Yes/No) and whether participants would consider themselves physically active (Yes/No), and if ‘Yes’, their reported average length of time (in minutes) per exercise session each day was used to derive the categories (adequate: at least 30 min and inadequate: <30 min).

The final data set allowed for identification of any trend between the outcome and exploratory variables. 

### 2.6. Statistical Analysis

Preliminary analyses included frequency tabulations of all selected characteristics for each of the study periods (2001–2003 and 2016–2018), and this was followed by trends in the prevalence of obesity over a 15-year period. The differences in the prevalence of obesity over the two survey years were estimated using a chi-squared to test the significance of differences at *p* < 0.05. Bivariate logistic regression analyses were used to examine the unadjusted odds ratio and multiple logistic regression analyses were used to examine factors associated with obesity. In this study, a staged modelling method was conducted for the multivariable analysis in which exploratory variables were entered in stages into the model to assess factors associated with the study outcome [18]. First, the demographic factors were entered into the baseline multivariable model to examine associated factors with obesity. Thereafter, a manual stepwise backwards elimination process was conducted and only variables significantly associated with the study outcome at *p*-values < 0.05 were retained in the model (model 1). Second, socio-economic factors were entered into model 1, and those factors with *p*-values < 0.05 were retained (model 2) after a backwards elimination process was conducted. Third, dietary factors were added to model 2. As we did previously, those factors with *p*-values < 0.05 were retained (model 3). Similarly, progressively modelling was carried for fourth (cooking method factors) and fifth (lifestyle factors), respectively, and those factors with *p*-values < 0.05 were retained and regarded as factors associated with obesity in rural Victoria. The odds ratios (OR) and their 95% confidence interval (CI) derived from the adjusted logistic regression models were used to measure the level of association of the factors with obesity and all analyses were performed using STATA version 17.0 (Stata Corporation, College Station, TX, USA).

### 2.7. Ethics Approval & Consent

The Goulburn Valley Health Research Ethics Committee granted approval for the Crossroads study II in May 2016 (GVH-20/16) and the Crossroads I study in 2001 (GCH3-/99). All participants of the household survey aged 16 years and older were provided with information on the study and gave written consent to participate. Separate consent forms were used for some additional elements of the clinical assessment study [12]. 

## 3. Results

Table 1 presents the characteristics of the participants in both Crossroads studies. Of the 7907 participants aged 18 years and over, who were included in the analyses, majority were older in Crossroads II (>55 years old 53.1%), had more female than male participants in both Crossroads studies (Crossroads I: 54.1% vs. 45.9%; Crossroads II: 57.7% vs. 42.3%) and more than 80% were born in Australia. Crossroads II represented a more even divide between rural and regional participants in contrast to Crossroads I which was predominated by regional residents (33.1% vs. 66.9%) and had more unemployed people. There were twice as many people in Crossroads II than Crossroads I with a tertiary education qualification (46.6% vs. 22.2%) and the proportion of married participants was about three times greater in Crossroads II (55.4% vs. 18.3%) 

In terms of dietary intake and lifestyle factors, the survey found that most participants had inadequate intake of the recommended serves of vegetables and dairy, which remained relatively unchanged across both studies. There were improvements in adequate fruit consumption, as well as a reduction in take-away foods between the two surveys. However, an increase in frying foods prepared at home, including eggs, chicken and meat was noted in Crossroads II. A greater number of people had a BMI of ≥30 kg/m^2^ in Crossroads II despite more people reporting adequate physical activity. 

### 3.1. Trends in Obesity Prevalence

Our results found a decrease in obesity prevalence among rural Victorians (i.e., from 27.7–24.8%) over the study period (Figure 1) which translates to a significant decrease of about 10.4% in 2016–2018. On the other hand, the prevalence of central obesity increased by about 11.1% over 15 years, particularly among women where greater changes in average waist circumference were observed (women: from 89.6 cm/35.3 inches to 94.8 cm/37.8 inches, *p* < 0.001; men: from 101.0 cm/39.8 inches to 102.8 cm/40.5 inches, *p* = 0.037) from 2001–2003 to 2016–2018, respectively.

### 3.2. Trends in Obesity Prevalence by Key Factors

Table 2 shows the trends in obesity prevalence by the demographic, socio-economic, dietary, cooking and lifestyle factors in this study. Obesity prevalence decreased by about 20.8% among rural dwellers but increased by about 64.0% among married people between 2001–2003 and 2016–2018. There was a trend towards an increase in obesity prevalence among participants who worked either full time (29.7%, *p* = 0.058) or part time (43.6%, *p* = 0.057) but this did not reach significance. The trend in obesity prevalence reduced significantly by approximately 6.1% and 26.0% among participants who consumed the recommended fruit intake of two serves a day, and those who do not drink alcohol, respectively. Comparatively between the two surveys, significant increases in obesity prevalence occurred in those who used frying as a method of cooking meat, chicken, and eggs, as opposed to those who use other methods of cooking with increases of about 53.5%, 31.4% and 58.6 respectively. 

### 3.3. Unadjusted and Adjusted Odd Ratios of Factors Associated with Obesity

Table 3 presents unadjusted and adjusted odd ratios for the factors associated with obesity across both Crossroads I and II surveys, the 95% CI and their levels of significance. The odds for obesity were similar between the Crossroad studies after adjusting for potential covariates (aOR 0.93: 95% CI 0.78, 1.10), but older participants aged 35–54 years (aOR 1.81: 95% CI 1.42, 2.31) and 55 years and older (aOR 1.96: 95% CI 1.56, 2.46) had higher odds for obesity compared with younger participants (18–34 years) (Table 3). Significantly higher odds for obesity was observed among participants who had no work at the time of the Crossroad studies (OR 1.29: 95% CI 1.01, 1.66) compared with those who worked full time, but this was nullified after adjusting for potential confounders. The odds for obesity were significantly lower (OR 0.77: 95% CI 0.63, 0.94 and OR 0.65: 95% CI 0.50, 0.85) compared with those who completed secondary education or lower and married people, respectively, but these were lost after adjustment. 

In terms of dietary intake, those who used fat-based spreads reported significantly higher odds of obesity (aOR 1.26: 95% CI 1.07, 1.48) compared with those who did not use fat-based spreads in their diet. A decreased risk of obesity was also noted in those who had inadequate (aOR 0.80: 95% CI 0.65, 0.97) and adequate physical activity (aOR 0.69: 95% CI 0.58, 0.82) compared with those who had no physical activity (Table 3).

### 3.4. Relationship between Obesity Prevalence and Fruit and Vegetable Consumption 

Across both surveys, obesity prevalence remained consistently high among those who did not consume fruits and vegetables. Having consumed either fruit or vegetables slightly lowered the risk of obesity among participants, while consuming both fruits and vegetables further lowered the risk of obesity overall (Figure 2). 

## 4. Discussion

This study investigated the trends and factors associated with obesity in the Goulburn Valley of Victoria, Australia, using data from the two Crossroads studies conducted 15 years apart. These data allowed for a unique opportunity to observe diet, other lifestyle, and health changes over an extended period, providing time to show evolution and developments within the community and look at any reflected effect. This study found that obesity prevalence increased significantly over 15 years, more among those who ate fried food, but decreased significantly among rural dwellers and those who had adequate fruit intake. The increase in obesity prevalence was significantly associated with older age (≥35 years), use of fat-based spreads for breads and physical inactivity. The impact of the various factors including demographic, socioeconomic, dietary, cooking method and lifestyle factors on obesity prevalence among the study participants have been discussed in their respective sub-sections below.

### 4.1. Demographics

Of the present study, one unexpected finding in terms of demographics was that the reduced prevalence of obesity among people living in the smaller, rural towns after 15 years. In an area that is renowned for farming fresh foods, it is hypothesized that perhaps the accessibility to fresh produce is in some ways better in the rural towns. Rural towns are also less likely to have many take-away stores and nearby supermarkets selling an array of discretionary food items, whereas in the regional centers, there is often greater exposure and accessibility to purchase discretionary foods, particularly for the purpose of convenience. There may also be an influence of lifestyle and the kind of jobs that are more common in rural areas, such as farming and manual labor. Interestingly, our findings were consistent with those of the Australian Institute of Health and Welfare (AIHW) in 2017–2018 [19,20], which suggested that the prevalence of overweight and obesity is slightly lower in outer regional/rural and remote areas compared to inner regional centers, although overweight and obesity prevalence is still on the rise. Another Australian study also found that those living in rural areas were less likely to be living with obesity than those in regional centers [21]. 

This study also found that the risk of obesity was higher among those who were aged 35 years and older, compared to people aged 18–34 years. While obesity continues to grow globally across all age groups, research has shown the prevalence steadily increases with age after 20 years old, with peak obesity being reached at age group 50–65 [22]. After age 65, a small decrease in obesity was noted in previous research [22], a trend that highlights the importance for age tailored exercise Programs and diet education to be accessible to members of the community to promote healthy lifestyles that assist in reducing any increased risks of obesity with age. 

Whilst there was evidence of decreased prevalence of general obesity, there was also evidence of increased prevalence of central obesity. A significant increase in waist circumference, particularly in women, between the two Crossroads studies raises an interesting question in the difference between the lifestyle determinants in men and women. Janus et al. [23] reflected a similar finding in their study conducted across rural Victorian and South Australian towns, showing a greater prevalence of central obesity in the surveyed women compared with men. Research suggests that waist circumference is associated with a diet high in energy dense foods with a high glycemic index [24]. Future research into daily dietary intake, such as 24-h dietary recalls, or food frequency questionnaires, would give a greater insight into this finding and assist in developing proactive policy to improve health outcomes in the region.

### 4.2. Socio-Economic Factors

Socio-economic factors accounted for a small proportion of the factors associated with obesity in this study. Increased obesity prevalence was found among those who reported not working, as opposed to those engaged in part-time or full-time employment. However, after adjusting for the potential confounders, this significance was lost. This was also the case for the reduced prevalence of obesity which was observed among those who reported never being married and those with university education. These results indicate that within the Goulburn Valley community, obesity prevalence is influenced more by diet, lifestyle, and demographic factors than socio-economic factors. 

### 4.3. Dietary Factors

The study observed a lower prevalence of obesity among those who had adequate daily fruit intake, alongside an encouraging improvement in adequate daily fruit consumption in the 15 years between surveys. However, adequate daily vegetable consumption had decreased over this same period. The results show that those who consumed no fruit or vegetable had a higher prevalence of obesity than those who consumed fruits, vegetables, or both. Whilst health and nutrition information has become increasingly accessible, it would appear many of the established, less than optimal dietary practices in the population have not been impacted by nutrition, lifestyle, and diet awareness campaigns. Malnutrition in all forms, including obesity, has been linked to food insecurity [25] with a strong and positive association reported particularly among women, which was altered by marital status and various stressors [26]. The occurrence of food insecurity and the social determinants of health within rural and regional areas is not uncommon, where healthy food selections are frequently not accessible [27] and often more expensive [28], leaving discretionary food options a more convenient choice [23]. In a study conducted by Love et al. [17], food insecurity was found to be largely related to the higher price of healthy foods and lower cost of discretionary food items in rural towns, alongside inaccessibility. These factors have influence on the prevalence of obesity and chronic illnesses across rural regions [17]. Research into the link between nutrition literacy and improved dietary practice [29,30] suggested a correlation between insufficient education or resources regarding healthy food choices and poor dietary habits. One study reviewed found a direct association between good health literacy and increased fruit and vegetable intake [16]. Further investigation is needed to understand if health and nutrition literacy Programs are reaching regional and rural areas, and if so, identifying what barriers exist in impacting change in behavior. Health promotion strategies to improve vegetable intake and develop healthier cooking methods in the home should be considered to assist in health outcomes for the region.

### 4.4. Cooking Method Factors

Participants reported increases in using frying as a method of cooking meat, chicken and eggs at home between the two Crossroads studies. The use of frying as a method of cooking these items was significantly associated with obesity prevalence. In addition, participants who indicated that they consume fat-based spreads showed higher odds of obesity. Previous reports have linked excess caloric intake to chronic diseases [15,31]. In one study, overconsumption of fats increased the risk of obesity, coronary heart disease, diabetes, and cancer [31] and in another, a better quality diet resulted in better quality of life and greater emotional wellbeing, especially among women [32]. There is a need for greater focus on addressing the diet quality in population Programs and policy that support wellbeing and healthy ageing, where household food security can be achieved. 

### 4.5. Lifestyle Factors

There was a significant association between obesity and exercise, such that increased participation in physical activity is more likely to reduce the obesity prevalence among the participants in this study. This occurred even when the participants did not meet the recommended exercise guidelines. Therefore, it is crucial to design sustainable strategies encouraging participation in any form of physical activity. Such strategies may include development of targeted Programs that increase awareness about the benefits of physical activity. This can be done by providing relevant information in community newsletters, community groups, and social media. People also need assistance in identifying physical activity options that match their interests, lifestyles, and functional abilities and identify opportunities for them to pursue these activities. This finding further highlights the need for Programs or resources describing physical activity opportunities for different age groups, which could be provided to health practitioners to make appropriate referrals and recommendation when needed [33]. 

The results of the Crossroads II survey showed a slightly higher percentage of participants reported no alcohol intake compared to Crossroads I and a trend towards lower obesity prevalence in those who did not drink when comparing the two surveys. However, when looking at the odd ratios for factors associated with obesity, consuming alcohol showed reduced likelihood for obesity in the unadjusted state. When adjusted for potential confounders, this finding was nullified. Previous studies have shown that those who consume low risk amounts of alcohol, or light drinkers, have a reduced prevalence of obesity than those who are considered heavy drinkers, or those who do not drink alcohol [34,35]. However, the relationship between alcohol intake and obesity is multi-factorial and could be influenced by genetics [36] and various lifestyle factors [35].

### 4.6. Limitations and Strengths

The dietary questions asked across the household and clinic surveys provided a spread of answers which did not reflect benchmark Australian recommended daily intakes (RDI), thus not allowing a model comparison of specifics to RDI recommendations. Total energy intake was not assessed. As the data is cross-sectional, it only provides a snapshot of the community at the time of the surveys. In addition, income was not analyzed in this study, which is another important index of socio-economic status. The most significant amount of missing data from the survey responses was observed in the lifestyle section of the Crossroads II household survey, reporting up to 12% missing data [8]. Despite these limitations, this study has numerous strengths including its repeated cross-sectional nature incorporating cohorts from both rural towns and the regional center for comparison. The significant period between surveys allowed for a comparison of lifestyle and diet trends and habits that exist within the community. The design of this study allowed for an insight into changes over 15 years, a period where information and awareness into healthy eating practices has become more prevalent and accessible, and access to healthcare improved. Another strength of this study was the use of trained research assistants and professionals for data collection [8] and the robust analysis conducted, to improve the internal validity and minimize the influence of potential confounders on the study findings. The results of this study hold a valuable place within the current research into obesity in rural areas and demonstrate a need for further research into the success of healthy dietary Programs and policy implemented within rural communities, as well as their efficacy in prompting positive change to dietary behaviors. Research into the link between nutrition literacy and improved dietary practice [29,30] suggests a correlation between insufficient education or resources regarding healthy food choices and poor dietary habits. The results of the study also raise the question of healthy food accessibility within the area. Further, these findings remain consistent with several of the other studies reviewed, where overall insufficient dietary habits and nutrient intake was identified. 

## 5. Conclusions

This study found an overall increase in obesity prevalence across the Goulburn Valley region of Victoria, associated with an increase in unhealthy dietary behavior over time. Although there was a decline in obesity prevalence among rural dwellers, many modifiable dietary and lifestyle risk factors remained unchanged over the 15 years. It is acknowledged that the sample was comprised of a larger proportion of females (54.1% Crossroads-I, 57.7% Crossroads-II) compared with Australian national percentage of 50.2% (ABS, 2021) and older mean age (46 years Crossroads-I, 53 years Crossroads-II compared with 38 years (ABS, 2021) than the Australian population. While the cross-sectional nature of this study means its results may not represent the broader rural population, the reported findings hold important implications to public health policy and promotion within rural health and navigate areas of further investigation in other communities. The results also identified target groups and specific behaviour change for health promotion. The knowledge of the dietary and lifestyle risk factors identified would assist in creating targeted health policies that are better suited to the environment of the community and may prove to be more effective in assisting residents to make positive, healthy changes. Understanding and addressing the upstream determinants of dietary intake and food choices within the Goulburn Valley communities would also assist in the development of future health promotion Programs to reduce the impact of the obesity epidemic. We recommend further research into the scope and efficacy of nutrition and health literacy within the region, as well as further research into the determinants of dietary intake, to improve and assist in the changing of long-term food and lifestyle behavior changes. 

## Figures and Tables

**Figure 1 nutrients-14-04557-f001:**
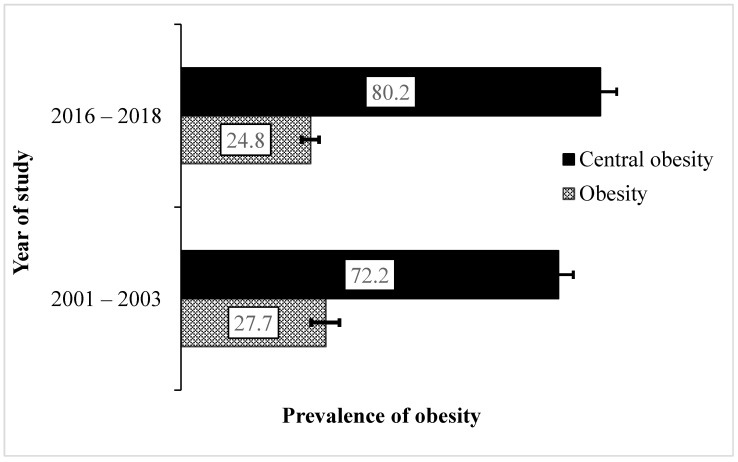
Trends in prevalence (%) and 95% confidence interval of obesity in the smaller towns surveyed in Victoria (2001–2018). General and central obesity measures were determined by body mass index and waist circumference, respectively.

**Figure 2 nutrients-14-04557-f002:**
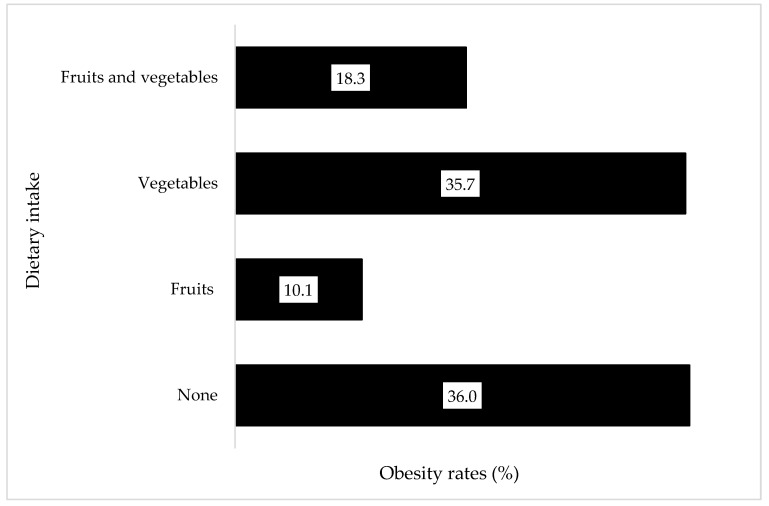
Obesity prevalence for any fruit or vegetable consumption across both Crossroads studies (2003–2018) Prevalence of obesity in those who consumed no fruit or vegetable, those who consumed any fruit or any vegetable, and those who consumed both fruits and vegetables are shown.

**Table 1 nutrients-14-04557-t001:** Characteristics of the study sample in Crossroads 1 (2001–2003) and II (2016–2018).

Variables	2001–2003 (*N* = 5258), *n* (%)	2016–2018 (*N* = 2649), *n* (%)
Demographic Factors		
Age groups		
18–34 years	1535 (30.2)	550 (20.8)
35–54 years	1840 (36.2)	691 (26.1)
55 and over	1703 (33.5)	1407 (53.1)
Sex		
Male	2439 (45.9)	1128 (42.3)
Female	2879 (54.1)	1539 (57.7)
Ethnicity (country of birth)		
Australian born	4215 (88.4)	2243 (84.1)
Non- Australian born	551 (11.6)	423 (15.9)
Place of residence		
Rural	1762 (33.1)	1336 (49.9)
Regional	3566 (66.9)	1344 (50.2)
Socio-Economic Factors		
Working status		
Working full time	1719 (37.2)	216 (43.0)
Working part time	799 (17.3)	130 (25.9)
Not working	2108 (45.6)	156 (31.1)
Educational status		
Completed secondary or less	3704 (77.9)	1422 (53.4)
Completed TAFE, Certificate or Diploma	439 (9.2)	700 (26.3)
Completed University	615 (12.9)	539 (20.3)
Marital status		
Married/defacto	352 (18.3)	1405 (55.3)
Divorced/widowed/separated	794 (41.2)	759 (29.9)
Never married	782 (40.6)	376 (14.8)
Dietary Factors		
Daily vegetable intake		
Adequate	1319 (27.5)	671 (26.5)
Inadequate	3485 (72.5)	1858 (73.5)
Daily fruit intake		
Adequate	2550 (53.0)	1442 (57.0)
Inadequate	2258 (47.0)	1087 (43.0)
Daily dairy intake		
Adequate	1641 (34.1)	970 (38.4)
Inadequate	3174 (65.9)	1559 (61.6)
Daily take-away consumption		
Yes	1923 (40.0)	920 (36.4)
No	2886 (60.0)	1610 (63.6)
Use of fat-based spreads on bread		
Yes	1383 (29.2)	768 (30.4)
No	3359 (70.8)	1760 (69.6)
Cooking method Factors		
How meat was usually cooked		
Fried	173 (16.6)	183 (30.0)
Other	871 (83.4)	427 (70.0)
How egg was usually cooked		
Fried	275 (26.3)	236 (32.6)
Other	769 (73.7)	488 (67.4)
How chicken was usually cooked		
Fried	173 (16.6)	126 (17.5)
Other	872 (83.4)	593 (82.5)
Lifestyle Factors		
Body mass index classification		
Normal/underweight (<25.0 kg/m²)	324 (31.6)	710 (32.9)
Overweight (25.0–29.9 kg/m²)	413 (40.3)	784 (36.3)
Obese (≥30.0 kg/m²)	289 (28.2)	665 (30.8)
Smoking (current or past)		
Yes	1147 (23.7)	403 (15.9)
No	3696 (76.3)	2125 (84.1)
Drinking alcohol (current)		
Yes	3124 (64.5)	1611 (63.7)
No	1721 (35.5)	919 (36.3)
Physical activity		
None	1643 (34.3)	790 (31.2)
Adequate	1938 (40.5)	1135 (44.9)
Inadequate	1206 (25.2)	605 (23.9)

**Table 2 nutrients-14-04557-t002:** Prevalence and differences as percentage-points of obesity prevalence by demographic, socio-economic, lifestyle and dietary characteristics in the Crossroads studies. Significant differences (*p* < 0.05) are indicated as Asterix. TAFE = Technical and Further Education.

Characteristics	Crossroads I	Crossroads II	Crossroads (I–II)
Demographic Factors			
Age groups			
18–34 years	19.4 (13.7, 26.9)	16.0 (13.7, 26.9)	3.4 (−3.8, 10.7)
35–54 years	28.8 (24.7, 33.3)	26.5 (24.7, 33.3)	2.4 (−3.0, 7.8)
≥55 years	29.9 (25.7, 34.4)	27.7 (25.7, 34.4)	2.1 (−2.8, 7.1)
Sex			
Male	26.9 (23.0, 31.1)	24.4 (22.0, 27.0)	2.5 (−2.2, 7.3)
Female	28.4 (24.9, 32.2)	25.2 (23.1, 27.4)	3.2 (−1.1, 7.5)
Ethnicity (country of birth)			
Australian born	27.7 (24.9, 30.7)	25.5 (23.7, 27.3)	2.3 (−1.1, 5.7)
Non-Australian born	27.2 (20.1, 35.7)	22.0 (18.3, 26.2)	5.2 (−3.5, 14.0)
Residency type			I
Rural	31.7 (27.0, 36.8)	25.1 (22.9, 27.5)	6.6 (1.2, 12.0) *
Regional	25.7 (22.6, 29.1)	24.5 (22.3, 26.9)	1.2 (−2.7, 5.2)
Socio-Economic Factors			
Working status			
Working full time	24.6 (20.5, 29.2)	31.9 (26.1, 38.5)	−7.3 (−14.9, 0.3)
Working part time	22.5 (16.9, 29.2)	32.3 (24.8, 40.8)	−9.8 (−20.0, 0.3)
Not working	32.7 (28.6, 37.1)	32.7 (25.8, 40.5)	−0.0 (−8.5, 8.5)
Educational status			
Completed secondary or less	28.8 (25.8, 32.1)	25.5 (23.3, 27.9)	3.3 (−0.6, 7.2)
Completed TAFE, Certificate or Diploma	25.3 (17.1, 35.7)	26.6 (23.4, 30.0)	−1.3 (−11.2, 8.6)
Completed University	23.7 (17.9, 30.7)	21.3 (18.1, 25.0)	2.3 (−5.0, 9.6)
Marital status			
Married/defacto	16.7 (9.4, 27.7)	27.4 (25.1, 29.8)	−10.7 (−20.0, −1.4) *
Divorced/widowed/separated	31.2 (25.0, 38.2)	27.3 (24.2, 30.6)	3.9 (−3.4, 11.3)
Never married	24.4 (16.1, 35.1)	18.4 (14.8, 22.6)	6.0 (−4.3, 16.3)
Dietary Factors			
Daily vegetable intake			
Adequate	29.2 (24.5, 34.4)	25.9 (22.8, 29.4)	3.3 (−2.7, 9.2)
Inadequate	27.1 (24.0, 30.5)	26.0 (24.1, 28.0)	1.1 (−2.7, 4.9)
Daily fruit intake			
Adequate	28.5 (25.0, 32.3)	23.9 (21.8, 26.2)	4.6 (0.3, 8.9) *
Inadequate	26.7 (22.9, 31.0)	28.7 (26.1, 31.5)	−2.0 (−6.8, 2.9)
Daily dairy intake			
Adequate	25.7 (21.5, 30.4)	27.4 (24.7, 30.3)	−1.7 (−6.9, 3.6)
Inadequate	28.8 (25.5, 32.4)	25.1 (23.0, 27.3)	3.7 (−0.3, 7.8)
Daily take-away consumption			
Yes	28.4 (24.5, 32.6)	23.8 (21.2, 26.7)	4.6 (−0.4, 9.5)
No	27.2 (23.7, 31.0)	27.2 (25.1, 29.4)	−0.02 (−4.3, 4.2)
Use of fat-based spreads on bread			
Yes	31.3 (26.5, 36.4)	28.1 (25.1, 31.4)	3.1 (−2.8, 9.0)
No	26.0 (22.9, 29.4)	25.1 (23.1, 27.1)	1.0 (−2.9, 4.8)
Cooking method Factors			
How meat was usually cooked			
Fried	22.1 (16.5, 28.9)	33.9 (27.4, 41.1)	−11.8 (−21.0, −2.5) *
Other	28.8 (25.9, 31.9)	30.9 (26.7, 35.5)	−2.1 (−7.4, 3.2)
How egg was usually cooked			
Fried	27.1 (22.2, 32.7)	35.6 (29.7, 41.9)	−8.5 (−16.6, −0.4) *
Other	27.9 (24.9, 31.2)	30.1 (26.2, 34.4)	−2.2 (−7.4, 3.0)
How chicken was usually cooked			
Fried	21.5 (16.0, 28.3)	34.1 (26.4, 42.8)	−12.6 (−22.9, −2.3) *
Other	29.0 (26.1, 32.1)	30.9 (27.3, 34.7)	−1.8 (−6.6, 3.0)
Lifestyle Factors			
Smoking			
No	27.9 (25.0, 31.1)	26.6 (24.8, 28.6)	1.3 (−2.3, 4.9)
Yes	26.9 (21.2, 33.4)	22.6 (18.8, 26.9)	4.3 (−3.1, 11.7)
Drinking alcohol			
No	35.4 (30.4, 40.9)	26.2 (23.5, 29.2)	9.2 (3.2, 15.2) *
Yes	24.3 (21.3, 27.5)	25.8 (23.7, 28.0)	−1.6 (−5.3, 2.2)
Physical activity			
None	33.7 (28.6, 39.1)	29.2 (26.2, 32.5)	4.4 (−1.7, 10.5)
Inadequate	23.6 (18.6, 29.3)	26.9 (23.6, 30.6)	−3.4 (−9.8, 3.0)
Adequate	26.0 (22.3, 30.1)	23.2 (20.8, 25.7)	2.8 (−1.8, 7.4)

**Table 3 nutrients-14-04557-t003:** Adjusted and unadjusted odd ratios with 95% CI for factors associated with obesity in regional Victoria (Crossroads studies 2001–2003 and 2015–2018). TAFE = Technical and Further Education.

Variables	Unadjusted OR (95% CI)	*p*-Value	Adjusted OR (95% CI)	*p*-Value
Year of study				
2001–2003	Ref		Ref	
2016–2018	0.86 (0.73, 1.01)	0.067	0.93 (0.78, 1.10)	0.368
Demography				
Age				
18–34 years	Ref		Ref	
35–54 years	1.88 (1.48, 2.39)	<0.001	1.81 (1.42, 2.31)	<0.001
55 and over	1.96 (1.57, 2.46)	<0.001	1.96 (1.56, 2.46)	<0.001
Sex				
Male	Ref			
Female	1.05 (0.91, 1.22)	0.501	-	
Ethnicity (country of birth)				
Australian Born	Ref			
Non-Australian Born	1.17 (0.95, 1.45)	0.146	-	
Place of residence				
Rural	Ref			
Regional	0.92 (0.79, 1.07)	0.263	-	
Socio-Economic Factors				
Working status				
Full time	Ref			
Part time	0.97 (0.71, 1.32)	0.835	-	
Not working	1.29 (1.01, 1.66)	0.040	-	
Educational status				
Completed secondary or less	Ref			
Completed TAFE, Certificate or Diploma	0.99 (0.82, 1.19)	0.886	-	
Completed University	0.77 (0.63, 0.94)	0.011	-	
Marital status				
Married, defacto	Ref			
Divorced, widowed, separated	1.06 (0.88, 1.27)	0.540	-	
Never married	0.65 (0.5, 0.85)	0.001	-	
Dietary Factors				
Daily vegetable intake				
Adequate	Ref			
Inadequate	1.04 (0.88, 1.22)	0.677	-	
Daily fruit intake				
Adequate	Ref			
Inadequate	0.86 (0.74, 1.00)	0.053	-	
Daily dairy intake				
Adequate	Ref			
Inadequate	1.04 (0.89, 1.21)	0.622	-	
Daily take-away consumption				
No	Ref			
Yes	0.91 (0.78, 1.06)	0.220	-	
Use of fat-based spreads on bread				
No	Ref		Ref	
Yes	1.21 (1.03, 1.42)	0.019	1.26 (1.07, 1.48)	0.005
Cooking method Factors				
How meat was usually cooked				
Fried	Ref			
Other	1.07 (0.82, 1.39)	0.620	-	
How egg was usually cooked				
Fried	Ref			
Other	0.90 (0.72, 1.12)	0.347	-	
How chicken was usually cooked				
Fried	Ref			
Other	1.16 (0.87, 1.53)	0.312	-	
Lifestyle Factors				
Smoking				
No	Ref			
Yes	0.85 (0.7, 1.05)	0.129	-	
Drinking alcohol				
No	Ref			
Yes	0.85 (0.73, 0.99)	0.037	-	
Physical Activity				
None	Ref		Ref	
Inadequate	0.80 (0.65, 0.98)	0.028	0.80 (0.65, 0.97)	0.027
Adequate	0.72 (0.61, 0.85)	<0.001	0.69 (0.58, 0.82)	<0.001

## Data Availability

Data supporting reported results have been presented and additional raw data can be obtained from the corresponding author on reasonable request.

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
