# Peer review of "Trends and Factors Associated with Obesity Prevalence in Rural Australian Adults—Comparative Analysis of the Crossroads Studies in Victoria over 15 Years"

_nutrients, 2022, doi:10.3390/nu14214557_

Round 1

Reviewer 1 Report

Manuscript ID: nutrients-1909202

Title: Trends and Factors Associated with Obesity Prevalence in Rural Australia - Comparative analysis of the Crossroads studies in Victoria over

15 years

Dear Editor thank you for sending me this manuscript to review

This manuscript reports on two data collections of the Crossroads study in which the time points were 15 years apart.

Overall, I enjoyed reading this paper as it was well-written and provides an important update to the literature on population level changes in obesity prevalence.

I have some minor comments to make:

Abstract. It would be nice if the authors could be clear that this study is a repeated cross-sectional survey. This is not clear until the end of the Introduction.

Main body. There is some inconsistency in cross sectional and cross-sectional throughout the manuscript.

The authors state that pregnant women were unable to participate. Could the authors please use more inclusive language ie pregnant people and also clarify why they were “unable” to participate. I think this may have been a poor choice of description by the authors. If pregnant people were excluded there needs to be a reason.

Section 2.5, line 196 States three years of the survey. Is this correct?

Given the study was conducted face-to-face response rates for both surveys need to be reported.

Table 1, I would have included the sample N for each survey year at the top of the table

Discussion. Please discuss the results that obesity prevalence decreased but central obesity did not over the two surveys.

I found the para beginning on line 340 to be confusing. The authors state the lack of statistical significance of the results, but this is not supported by Table 3. Can the authors please clarify their discussion points here? Were they talking about obesity prevalence or central obesity? Perhaps some sub-headings might help the reader navigate the Discussion.

Finally can the authors comment on how representative this sample is to the broader Australian population.

Author Response

Reviewer 1

Dear Editor thank you for sending me this manuscript to review

This manuscript reports on two data collections of the Crossroads study in which the time points were 15 years apart. Overall, I enjoyed reading this paper as it was well-written and provides an important update to the literature on population level changes in obesity prevalence. I have some minor comments to make:

Abstract.

  1. It would be nice if the authors could be clear that this study is a repeated cross-sectional survey. This is not clear until the end of the Introduction.

Response: Added to line 24.

  1. Main body. There is some inconsistency in cross sectional and cross-sectional throughout the manuscript.

Response: Changes made- Hyphenated ‘cross-sectional’ has been used across the manuscript.

  1. The authors state that pregnant women were unable to participate. Could the authors please use more inclusive language ie pregnant people and also clarify why they were “unable” to participate. I think this may have been a poor choice of description by the authors. If pregnant people were excluded there needs to be a reason.

Response: As part of the exclusion criteria during the Crossroads studies, data were not collected for pregnant people which could be due to the physiological changes during pregnancy. This has been clarified in the manuscript [see line 136]

  1. Section 2.5, line 196 States three years of the survey. Is this correct?

Response: This has been corrected in the statement below to reflect the study period.

‘Preliminary analyses included frequency tabulations of all selected characteristics for each of the study periods [2001-03 and 2016-18].- [Added to line 200]

  1. Given the study was conducted face-to-face response rates for both surveys need to be reported.

Response: Added to text- line 140-143, and now reads:

‘Overall, 70.3% (n=5258) of the invited households in Crossroads I responded to the survey and 61.3% of respondents attended the clinical assessment(1). In Crossroads II, the response rate was 62.7% (n= 2649) out of the 3022 households that were invited to participate in the survey, with 48.1% attending the clinical assessment’

  1. Table 1, I would have included the sample N for each survey year at the top of the table

Response: Added to table 1

Discussion.

  1. Please discuss the results that obesity prevalence decreased but central obesity did not over the two surveys.

Response: This has been discussed in a new paragraph. [see line 344]

‘Whilst there was evidence of decreased prevalence of obesity, there was evidence of increased prevalence of central obesity. A significant increase in waist circumference, particularly in women, between the two Crossroads studies raises an interesting question in the difference between the lifestyle determinants in men and women. Janus et al (2) reflected a similar finding in their study conducted across rural Victorian and South Australian towns, showing a greater prevalence of central obesity in the surveyed women compared with men. Research suggests that waist circumference is associated with a diet high in energy dense foods with a high glycemic index(3). Future research into daily dietary intake, such as 24-hour dietary recalls, or food frequency questionnaires, would give a greater insight into this finding and assist in developing proactive policy to improve health outcomes in the region’

  1. I found the para beginning on line 340 to be confusing. The authors state the lack of statistical significance of the results, but this is not supported by Table 3. Can the authors please clarify their discussion points here? Were they talking about obesity prevalence or central obesity? Perhaps some sub-headings might help the reader navigate the Discussion.

Response: For that comment, we meant that various aspects of diet and lifestyle practices, particularly with vegetable consumption, the use of fat-based spreads, alcohol consumption and physical activity did not change significantly between the Crossroads studies. However, the section has been revised due to suggestions from other reviewers. We have also now added sub-heading for easy comprehension and enhanced flow of the manuscript

  1. Finally can the authors comment on how representative this sample is to the broader Australian population.

Response: The comment below was added to reflect the issue of generalizability of the results in the conclusion

‘It is acknowledged that the sample was comprised of a larger proportion of females (54.1% Crossroads-I, 57.7% Crossroads-II) compared with Australian national percentage of 50.2% (ABS, 2021) and older mean age (46 years Crossroads-I, 53 years Crossroads-II compared with 38 years (ABS, 2021) than the Australian population. While the cross-sectional nature of this study means its results may not represent the broader rural population, the reported findings hold important implications to public health policy and promotion within rural health and navigate areas of further investigation in other communities.’

Thank you for your feedback.

Reviewer 2 Report

Nice work

Maybe I missed it, but I am confused by the Appendix A and B.  Should the text associated with A be deleted?  And should Table A1 be in the Appendix A section?

Author Response

Reviewer 2

Comments and Suggestions for Authors

Nice work

  1. Maybe I missed it, but I am confused by the Appendix A and B.  Should the text associated with A be deleted?  And should Table A1 be in the Appendix A section?

Response: Thank you for bringing this to our attention. This has been amended and Table 1A has been renamed Supplementary Table 1 as suggested by the previous reviewer and submitted as a separate document [supplementary document].

Reviewer 3 Report

Title of study: Define the study population in your study title – among adolescents, adults, and elderly.

Abstract: The title of study exhibiting the prevalence and trends of obesity but the beginning of abstract is showing about the dietary intake. I suggest you to begin the abstract with obesity rather than with dietary pattern.

Add with in line 36 after associated

Keywords: remove all keywords except obesity and choose keywords from your study title

Introduction: Sentence “Though WC  . . . tone and ethnicity” seem confusing and there is no reference. Make it simpler and add some reference

I suggest you delete the content related to type 2 diabetes, because it is creating confusion.

I have thoroughly read the introduction, and I feel that you need to rewrite your introduction. Most of the content of your introduction should be part of methodology section. Create a new heading study design and study datasets. Then discuss your study design first and after that explain the datasets which you used in your study. While writing a manuscript we need to give an overview of the dataset.

The paragraph related to Victoria should be removed from the manuscript.

Add some more paragraphs and add a good rationale with objectives at the end of manuscript.

Methods:

Study design: line 128 write ethics approval number

Keep study participant heading after study design. Because we want to know what is the study and where study was performed first then we want to know to whom study was conducted

In line 155, also add waist circumference in inches.

Section 2.4 is very informative, I suggest you write all the five classes into five different bullets.

Results:

Figure 1: Can you label figure 1 with prevalence of obesity and central obesity. Because the text is confusing for me

Check sentence line 255 “ The were a trend . . . . “

Table 2: Remove the last column p-value rather add an asterik sign on all the significant variables and categories. Also explain that asterik sign at footnote.

Remove the line “In comparison to  . . . . nullified after adjustment”  282-284

Remove the line “similar results were noted  . . ..  never married” 275-276

Figure 2: Label each bar.

I suggest you present these categories in figure 1 (obesity and central obesity) instead of year

Also fruits-vegetables-both-none in figure 2

Discussion:

The first line is contradictory. The paper investigated the relationship with food consumption. If you look at results section, it is showing 5 factors: demographic, socioeconomic, lifestyle, food and cooking. Write all. . .

The line 300-302 should be part of methods. Because in discussion you do not need to introduce anything. Remove the sentence “ The crossroad ………. rural setting”

In line 326 . ..  defined the age group of obesity instead of writing older group . . .

In line number 345-348, you have written Obesity is linked with food insecurity. As far as I know, undernutrition is . . . You can write the statement malnutrition of any form including obesity is associated with food insecurity and add some more citations.

Limitations & strengths:

Discuss about the missing data and data quality either as strength or weakness

Move appendices into the supplementary file. It should not be kept as part of main manuscript

Author Response

Reviewer 3

Comments and Suggestions for Authors

  1. Title of study: Define the study population in your study title – among adolescents, adults, and elderly.

Response: Adults added to title.

  1. Abstract: The title of study exhibiting the prevalence and trends of obesity but the beginning of abstract is showing about the dietary intake. I suggest you to begin the abstract with obesity rather than with dietary pattern.

Response: the introductory section of the abstract has been revised to start with obesity. It now reads:

‘This study examined the changes in prevalence of obesity and associated lifestyle factors using data from repeated cross-sectional, self-reported surveys (Crossroads I: 2001-2003 and Crossroads II: 2016-2018, studies) and clinic anthropometric measurements collected from regional and rural towns in the Goulburn Valley, Victoria. Also, considering that past studies have only focused on the specifics of a community’s dietary intake categorically, or assessed caloric energy intake, we examined the difference in broad dietary practices over the study period’

  1. Add with in line 36 after associated

Response: Done.

  1. Keywords: remove all keywords except obesity and choose keywords from your study title

Response: Thanks for the comment. We have revised the keywords to include those in the title.

Introduction:

  1. Sentence “Though WC  . . . tone and ethnicity” seem confusing and there is no reference. Make it simpler and add some reference

Response: This section was deleted during review and correction no longer necessary.

  1. I suggest you delete the content related to type 2 diabetes, because it is creating confusion.

Response: Sentence relating to T2DM has been deleted.

  1. I have thoroughly read the introduction, and I feel that you need to rewrite your introduction. Most of the content of your introduction should be part of methodology section. Create a new heading study design and study datasets. Then discuss your study design first and after that explain the datasets which you used in your study. While writing a manuscript we need to give an overview of the dataset.

Response: Thanks for the comments. We have taken this onboard and the introduction has been restructured, the two new sub-headings were created in the methods section. The description of the dataset and study design are now included in the methods

  1. The paragraph related to Victoria should be removed from the manuscript.

Response: We think the reference to Victoria was necessary for setting context. However, this paragraph has been moved and included under the sub-heading ‘settings’ in the methods. We are happy to remove it if the reviewer thinks the positioning is not appropriate.

  1. Add some more paragraphs and add a good rationale with objectives at the end of manuscript.

Response: New information was added to give rationale in the introduction and at the end of the manuscript.

‘The findings of this study will provide insight into the factors that influence obesity prevalence within this rural Victorian community. In addition, the findings from this investigation hold the potential to guide future health policy development to promote positive diet and other lifestyle changes, particularly within regional and rural communities. It may also serve to steer the direction of future research within the field of obesity and dietary intake patterns’ [See line 85]

In the conclusion:

….the reported findings hold important implications to public health policy and promotion within rural health and navigate areas of further investigation in other communities. The results also identified target groups and specific behaviour change for health promotion. The knowledge of the dietary and lifestyle risk factors identified would assist in creating targeted health policies that are better suited to the environment of the community and may prove to be more effective in assisting residents to make positive, healthy changes. Understanding and addressing the upstream determinants of dietary intake and food choices within the Goulburn Valley communities would also assist in the development of future health promotion programs to reduce the impact of the obesity epidemic. [Added from line 461 onwards]

Methods:

  1. Study design: line 128 write ethics approval number

Response: Ethics approval number was provided under sub-heading 2.6. [See line 224]

  1. Keep study participant heading after study design. Because we want to know what is the study and where study was performed first then we want to know to whom study was conducted

Response: Done. Order of paragraphs was also revised in line with previous suggestion.

  1. In line 155, also add waist circumference in inches.

Response: Done across the manuscript

  1. Section 2.4 is very informative, I suggest you write all the five classes into five different bullets.

Response: Bullet points have been added

Results:

  1. Figure 1: Can you label figure 1 with prevalence of obesity and central obesity. Because the text is confusing for me

Response: Revised figure 1 has been provided with the labels

  1. Check sentence line 255 “ The were a trend . . . . “

Response: Corrected to “There was a trend” in line 263

  1. Table 2: Remove the last column p-value rather add an asterisk sign on all the significant variables and categories. Also explain that asterisk sign at footnote.

Response: P-value column was removed, and asterisk used to indicate significant differences. This was also explained in the legend

  1. Remove the line “In comparison to  . . . . nullified after adjustment”  282-284

Response: Sentence was removed, as suggested

  1. Remove the line “similar results were noted  . . ..  never married” 275-276

Response: Sentence removed, as suggested

  1. Figure 2: Label each bar.

Response: Done.

  1. I suggest you present these categories in figure 1 (obesity and central obesity) instead of year

Response: Done.

New figure has been provided to show the obesity and central obesity across both studies

Also fruits-vegetables-both-none in figure 2

Response: We have provided a revised figure as advised

Discussion:

  1. The first line is contradictory. The paper investigated the relationship with food consumption. If you look at results section, it is showing 5 factors: demographic, socioeconomic, lifestyle, food and cooking. Write all. . .

Response: First line has been revised to provide a summary of the findings. The section now reads:

‘This study investigated the trends and factors associated with obesity in the Goulburn Valley of Victoria, Australia, using data from the two Crossroads studies conducted 15 years apart. These data allowed for a unique opportunity to observe diet, other lifestyle, and health changes over an extended period, providing time to show evolution and developments within the community and look at any reflected effect. This study found that obesity prevalence increased significantly over 15 years, more among those who ate fried food, but decreased significantly among rural dwellers and those who had adequate fruit intake. The increase in obesity prevalence was significantly associated with older age (≥35years), use of fat-based spreads for breads and physical inactivity. The impact of the various factors including demographic, socioeconomic, dietary, cooking method and lifestyle factors on obesity prevalence among the study participants have been discussed in their respective sub-sections below’ [See line 307]

  1. The line 300-302 should be part of methods. Because in discussion you do not need to introduce anything. Remove the sentence “ The crossroad ………. rural setting”

Response: Removed sentence “The Crossroads studies remain the most recent and comprehensive repeated cross-sectional survey of its kind within a rural setting’

  1. In line 326 . ..  defined the age group of obesity instead of writing older group . . .

Response: Age groups defined [See line 336]

  1. In line number 345-348, you have written Obesity is linked with food insecurity. As far as I know, undernutrition is . . . You can write the statement malnutrition of any form including obesity is associated with food insecurity and add some more citations.

Response: Amended from line 375

Limitations & strengths:

  1. Discuss about the missing data and data quality either as strength or weakness

Response: Done. This has been discussed in the discussion under ‘Limitations and strengths’

  1. Move appendices into the supplementary file. It should not be kept as part of main manuscript

Response: Completed. New document created with supplementary table to be submitted with manuscript.

Thank you for your feedback.
